# L1CAM as an E-selectin Ligand in Colon Cancer

**DOI:** 10.3390/ijms21218286

**Published:** 2020-11-05

**Authors:** Fanny M. Deschepper, Roberta Zoppi, Martina Pirro, Paul J. Hensbergen, Fabio Dall’Olio, Maximillianos Kotsias, Richard A. Gardner, Daniel I.R. Spencer, Paula A. Videira

**Affiliations:** 1Unidade de Ciências Biomoleculares Aplicadas (UCIBIO), Departamento Ciências da Vida, Faculdade de Ciências e Tecnologia, Universidade Nova de Lisboa, 2829-516 Caparica, Portugal; f.deschepper@campus.fct.unl.pt (F.M.D.); r.zopppi@campus.fct.unl.pt (R.Z.); 2Center for Proteomics and Metabolomics, Leiden University Medical Center, 2300 RC Leiden, The Netherlands; M.Pirro@lumc.nl (M.P.); P.J.Hensbergen@lumc.nl (P.J.H.); 3Department of Experimental, Diagnostic and Specialty Medicine (DIMES), University of Bologna, 40138 Bologna, Italy; fabio.dallolio@unibo.it; 4Ludger Ltd., Culham Science Centre, Abingdon, Oxfordshire OX14 3EB, UK; maximilianos.kotsias@ludger.com (M.K.); richard.gardner@ludger.com (R.A.G.); Daniel.Spencer@ludger.com (D.I.R.S.); 5CDG & Allies - Professionals and Patient Associations International Network (CDG & Allies - PPAIN), 2829-516 Caparica, Portugal

**Keywords:** colorectal cancer, E-selectin ligand, L1CAM, sialyl Lewis X antigen

## Abstract

Metastasis is the main cause of death among colorectal cancer (CRC) patients. E-selectin and its carbohydrate ligands, including sialyl Lewis X (sLe^X^) antigen, are key players in the binding of circulating tumor cells to the endothelium, which is one of the major events leading to organ invasion. Nevertheless, the identity of the glycoprotein scaffolds presenting these glycans in CRC remains unclear. In this study, we firstly have characterized the glycoengineered cell line SW620 transfected with the fucosyltransferase 6 (*FUT6*) coding for the α1,3-fucosyltransferase 6 (FUT6), which is the main enzyme responsible for the synthesis of sLe^X^ in CRC. The SW620FUT6 cell line expressed high levels of sLe^X^ antigen and E-selectin ligands. Moreover, it displayed increased migration ability. E-selectin ligand glycoproteins were isolated from the SW620FUT6 cell line, identified by mass spectrometry, and validated by flow cytometry and Western blot (WB). The most prominent E-selectin ligand we identified was the neural cell adhesion molecule L1 (L1CAM). Previous studies have shown association of L1CAM with metastasis in cancer, thus the novel role as E-selectin counter-receptor contributes to understand the molecular mechanism involving L1CAM in metastasis formation.

## 1. Introduction

According to the World Health Organization, colorectal cancer (CRC) is the third most common cancer and the second cancer related cause of death worldwide, with an estimated incidence of over 1.8 million cases in 2018 [1]. The high cancer mortality is mainly due to the formation of metastasis, formed by the colonization of a distant organ or lymph nodes by tumor cells detached from the primary tumor site: this has been observed in approximately 20% of patients with CRC at first diagnosis [2,3]. The organs mostly affected by CRC metastasis are the liver and the lungs, which are a result of the hematogenous dissemination [4]. To colonize a secondary site, tumor cells follow different steps of the so-called metastasis cascade. Firstly, tumor cells undergo the epithelial–mesenchymal transition (EMT) program, where the expression of the adhesion molecules is downregulated leading to the formation of motile tumor cells [5]. The interaction between the malignant tumor cells and their surrounding microenvironment are important. In particular, the focal adhesion kinases (FAK) and integrins mediate cell adhesion to the extracellular matrix and link the signaling communication between the inside and the outside of the cells [6]. In addition, NF-κB activation stimulates the inflammatory response of the tumor microenvironment and promotes cancer by supporting immune modulations [7]. The degradation of the extracellular environment, caused by enhanced proteolysis carried out mainly by matrix metalloproteinases, proceeds the intravasation of tumor cells into the lumen of lymphatic or blood vessels. The invasion of distant organs is possible only if the circulating tumor cells (CTCs) succeed in interacting and binding to the endothelium of the vessels. Through the expression of selectin ligands and subsequent interaction with endothelial selectins, CTCs gain the ability to roll on activated endothelium. This interaction is crucial for the deceleration of the CTCs in the bloodstream. The tethering and rolling of CTCs are followed by firm adhesion via integrins allowing transendothelial migration and development of a secondary tumor, i.e., metastasis formation [8,9].

The selectin family is composed of three transmembrane proteins called L-, E- and P-selectin. Selectins are C-type lectins and expressed by leukocytes (L-selectin), platelets (P-selectin) and endothelium (E- and P-selectin). The E-selectin minimal binding tetrasaccharides are the sialyl Lewis X (sLe^X^; NeuAcα2,3Galβ1,4[Fucα1,3]GlcNAc-R) antigen and its isomer sialyl Lewis A (sLe^A^; NeuAcα2,3Galβ1,3[Fucα1,4]GlcNAc-R) antigen. Aberrant glycosylation and in particular the overexpression of sLe^X/A^ is typically observed during malignant transformation of different types of cancer [10,11]. In CRC, the main glycan alterations are an increase in sialylation and fucosylation, including an increase in sLe^X/A^ antigen [12]. The overexpression of sLe^X^ antigen in CRC is mostly due to altered expression of the enzymes involved in their biosynthesis [13,14,15]. In CRC, the bases of increased expression of sLe^X^ are complex and multifactorial, although the last step of its biosynthesis is mediated mainly by FUT6 [16]. The importance of E-selectin-ligand interactions in CRC metastasis formation has been shown in several studies. In fact, blocking E-selectin interaction with tumor cells prevents metastasis formation in CRC [17,18]. E-selectin binding is modulated not only by the structural nature of the terminal glycan (of sLe^X/A^), but also by the protein scaffold carrying such post-translational modifications [19,20,21,22]. In different cancers, some E-selectin ligands have been identified such as the hematopoietic cell E/L-selectin (HCELL) glycoform of CD44, cutaneous lymphocyte antigen (CLA) glycoform of PSGL1, ESL1, CD43 (CD43-E) and CD13 [23,24,25]. In colon cancer, the E-selectin ligands identified so far are HCELL [26], carcinoembryonic antigen (CEA) [27], podocalyxin-like protein (PCLP), LAMP-1 (lysosomal membrane glycoprotein-1) and LAMP-2 [28]. Although these E-selectin ligand protein scaffolds have been identified in CRC, our knowledge of the nature of the E-selectin binding proteins is incomplete. The identification of clinically relevant E-selectin binding glycoproteins could offer novel targeted therapeutics capable of circumventing metastasis.

The aim of this study was to further elucidate the role of E-selectin ligands in CRC and identify the glycan profile and the protein scaffolds involved in CRC cells overexpressing E-selectin ligands. Thus, we have shown that the SW620 cell line, derived from a lymph node metastasis of colon cancer, transfected with the *FUT6* cDNA overexpressed E-selectin ligands. Isolated E-selectin ligands protein scaffolds were identified by mass spectrometry. Several candidates were identified, and among those, the neural cell adhesion molecule L1 (L1CAM). L1CAM was found to be overexpressed in *FUT6* overexpressing cells and, after immunoprecipitation, showed E-selectin binding. This study identified for the first time L1CAM as an E-selectin ligand. This novel role can help to understand the L1CAM molecular mechanism in metastasis formation.

## 2. Results

### 2.1. Characterization of SW620 Colon Cancer Cell Line Overexpressing FUT6

SW620 colon cancer cell line was transfected with a FUT6 expression vector or an empty plasmid [29] and the obtained cell lines were hereafter named SW620FUT6 or SW620Mock. In the glycoengineered SW620 cell line, the gene expression of *FUT6* and other α1,3/4-fucosyltransferases (*FUTs*) was evaluated by RT-qPCR (Table 1). As expected, SW620FUT6 presented significant higher *FUT6* mRNA levels compared to the Mock transfected cell line. Among other *FUTs*, *FUT5* and *FUT3* mRNA expression level was found decreased in SW620FUT6 cells, with the expression values being extremely low, especially for the *FUT3* gene. Other *FUT* mRNA expression levels were not significantly affected by *FUT6* transfection (Table 1). *FUT9* was either not expressed or expressed at an undetectable level by the experimental procedure (data not shown).

The biosynthesis of sLe^X^ antigen involves several glycosyltransferases depicted in Figure 1A and *FUT6* gene expression has been shown to impact sLe^X^ antigen expression in the SW620 cell line [29]. Thus, to confirm that the FUT6-overexpressing cell line shows an increase in sLe^X^ expression, we extracted and analyzed by Western blot (WB) with HECA-452 monoclonal antibody (mAb) proteins from SW620Mock and SW620FUT6 cell lines. This mAb reacts with both sLe^X^ and sialyl Lewis A (sLe^A^) antigens [30,31]. Figure 1B shows no staining of proteins from SW620Mock cells while proteins from SW620FUT6 cells presented several bands between 75 and 245 kDa. Further analysis by flow cytometry allowed us to establish a more specific expression pattern for sLe^X^ antigen, sLe^A^ antigen and E-selectin ligands (Figure 1C,D). For this purpose, HECA-452 mAb for sLe^X/A^ antigens, anti-CD15s for sLe^X^ antigen, anti-CA19-9 for sLe^A^ antigen and E-selectin chimera (E-Ig) for E-selectin ligands were used. Figure 1C shows intense staining for sLe^X/A^ antigens (HECA-452), sLe^X^ antigen (anti-CD15s) and E-selectin ligands (E-Ig) in SW620FUT6 cells, significantly increased compared to SW620Mock cells. According to CA19-9 staining, the sLe^A^ expression level is low in both SW620 variants, and even lower in SW620FUT6 cells than in SW620Mock cells. As represented in Figure 1D, the FUT6/Mock MFI ratios was high for sLe^X^ antigen and E-selectin ligands, but neatly unchanged for sLe^A^ antigen. This is not surprising, considering that FUT6 was unable to catalyze sLe^A^ antigen biosynthesis.

### 2.2. N-glycan Profiles of FUT6 vs. Mock Transfected SW620 Cells

In order to obtain more information on the glycosylated structures of SW620 cells transfected with *FUT6* compared to Mock, we extracted membrane proteins from both cell lines and profiles of *N*-glycans released after PNGase F treatment were obtained by HILIC-UHPLC-MS/MS. The masses of *N*-glycans from the two cell lines ranged from *m/z*^1+^ 960 and to *m/z*^4+^ 980. The ions were singly to quadruply charged. In Figure 2, each identified glycan structure is represented with its attributed peak ID. Peak IDs are reported in Appendix A for SW620Mock and SW620FUT6 cells, respectively. The composition of the *N*-glycans was confirmed by MS^n^ fragmentation analysis and the identified Y- and B-ion fragments for each structure are shown in Appendix A. In total, 69 and 78 *N*-glycans were identified in SW620Mock and SW620FUT6 cells, respectively. Identified structures included hybrid type and mono- to penta-antennae branched complex type *N*-glycans. Among these structures, several blood group antigens were identified in both cell lines; one structure with H antigen (SW620Mock: #34, SW620FUT6: #37), four structures with Le^A/X^ antigens (Mock: #36, #44, #60, #66; FUT6: #40, #47, #64, #68), three structures with sLe^A/X^ antigens (Mock: #57, #60, #66; FUT6: #65, #64, #68) and one structure with two fucoses carried on the antenna of a tri-antennae core-fucosylated *N*-glycan without specific identified B-ion fragments to determine the type of antigen (Mock: #51, FUT6: #58).

Among the identified *N*-glycans, some isomeric structures were observed. For example, structure #50 in SW620Mock and structure #53 in SW620FUT6 have the same *m/z*^3+^ value of 815.35. The two structures differ in the position of the antennary fucose (Fuc). In the Mock cell line, the Fuc residue is on the non-sialylated antennae, while in the FUT6 cell line the Fuc residue is on the sialylated antennae, giving, respectively, Le and sLe antigen structures. This difference was identified in the B-ion fragments. Characteristic B-ions of the sLe structure 803.34 (H1N1F1S1), 1289.63 (H4N1S1F1) and 1330.38 (H3N2S1F1) were identified in the SW620FUT6 #53 *N*-glycan whereas they were absent in the SW620Mock #50 *N*-glycan. Such differences appeared on other isomeric structures, such as *N*-glycan #65 and #68 from Mock compared to *N*-glycan #76 from FUT6. The difference concerned the core position of the Fuc residue in Mock versus sLe antigen structure in FUT6 established by comparison of the ion fragment intensities. Similar differences were attributed to the following *N*-glycan structures (Mock compared to FUT6): #38 compared to #34, #45 compared to #48, #55 compared to #59 and #67 compared to #73 and #74. All these differences were consistent with an increase of sLe structures due to the transfection of *FUT6* in SW620 cells.

Other structural differences were identified for other isomeric structures such as *N*-glycans #18 and #22 from Mock cells compared to *N*-glycans #21 and #25 from FUT6 cells. In these cases, a bisecting *N*-acetylglucosamine (GlcNAc) for the mono-galactosylated *N*-glycan #18 was identified in Mock but not in the FUT6 *N*-glycan #21 and inversely a bisecting GlcNAc for di-galactosylated *N*-glycan #25 was identified in FUT6 but not in the Mock cell *N*-glycan #22. Other neutral *N*-glycans were identified only in Mock or FUT6 cells, comprising hybrid and complex *N*-glycan structures. Furthermore, *N*-glycan structures with four sialylated antennae were identified only in SW620FUT6 cells.

### 2.3. FUT6 Overexpression Increases Migration Ability in SW620 Cells

Cell migration capacity was evaluated using a scratch wound healing assay. As observed in Figure 3A, qualitative observation of pictures at day 4 indicated that wounds in SW620FUT6 cells healed better than in SW620Mock cells and, at day 6, healed completely whilst some of wound area with SW620Mock cells were still free of cells. Quantitative measurements, plotted in Figure 3B, confirmed that SW620FUT6 cells have significant faster healing than SW620Mock cells. Noteworthy, healing of the wound was not the consequence of cell proliferation from the margins, but from the migration of cells inside the wound area (Figure 3A). Thus, cells transfected with *FUT6*, with increased expression of sLe^X^ antigen, possess improved migration ability compared to Mock transfected cells.

### 2.4. SW620FUT6 Cell Line Present High Expression of E-selectin Ligands

The expression of E-selectin ligands on SW620Mock and SW620FUT6 cell lines was evaluated by flow cytometry and WB. Flow cytometry analysis of the two cell lines highlighted a higher expression of E-selectin ligands on SW620FUT6 cells compared to SW620Mock cells (Figure 1C). Since E-selectin ligands can only interact with E-selectin if they are expressed on the cell surface, membrane proteins from SW620Mock and SW620FUT6 cells were extracted. WB analysis of these membrane proteins confirmed the flow cytometry results. Indeed, SW620Mock membrane proteins did not show stained bands, whereas SW620FUT6 presented E-selectin ligands at high molecular weight. Three main bands were identified at ∼100 kDa, between 135 and 180 kDa and ∼245 kDa (Figure 4A), respectively. E-selectin ligands were then successfully immunoprecipitated (IP) from a membrane protein extract of SW620FUT6 cells (Figure 4B). The E-selectin ligands were successfully isolated as the sLe^X/A^ staining showed.

### 2.5. Identification of E-selectin Ligands from SW620FUT6

Immunoprecipitated E-selectin ligands were analyzed by LC–MS/MS after SDS-PAGE and in-gel tryptic digestion. Four independent immunoprecipitates were analyzed, resulting in the identification of in total 1066 proteins, of which 434 were present in at least two samples (data not shown). Since E-selectin ligands are glycosylated proteins, the list was reduced to 57 glycoproteins, based on UniProtKB database annotations (Appendix A). Thirteen glycoproteins from this list were only identified as E-selectin ligands in SW620FUT6 cells but not in SW620Mock cells (Table 2). Most of these are plasma membrane proteins, according to UniProtKB database annotation. The top four E-selectin ligands from this list (based on the total number of spectral counts) had high molecular weight in line with the WB results (Figure 4B). Hence, we decided to focus our further analyses on these four proteins: neural cell adhesion molecule L1 (L1CAM), integrin α6 (ITGA6), receptor-type tyrosine-protein phosphatase eta (PTPRJ) and integrin β1 (ITGB1).

### 2.6. L1CAM, Integrin α6 and Integrin β1 Are Expressed on the Surface of SW620Mock and SW620FUT6 Cells

To confirm the expression of the E-selectin ligands on the cell surface, flow cytometry with SW620Mock and SW620FUT6 cells was performed. Indeed, with these analyses, L1CAM, integrin α6 and integrin β1 were detected on the cell surface (Figure 5A,B). However, PTPRJ presented almost no staining on both cell lines (data not shown). Interestingly, L1CAM presented a higher expression level on SW620FUT6 cells when compared to SW620Mock cells (Figure 5B). This higher expression was also emphasized by the WB staining assay (Figure 5C). The expression level of integrin α6 and integrin β1 was similar between SW620Mock and SW620FUT6 cells (Figure 5B).

### 2.7. L1CAM Is an E-selectin Ligand in SW620FUT6 Cells

The results above demonstrated L1CAM on the cell surface of both SW620Mock and SW620FUT6 cells. To confirm that only L1CAM from SW620FUT6 cells is an E-selectin ligand, we performed immunoprecipitation of L1CAM from these cell lines. Firstly, immunoprecipitation of L1CAM from SW620Mock and SW620FUT6 membrane proteins were successfully stained by mAb against L1CAM with a band around 245 kDa. Secondly, this L1CAM staining on immunoprecipitation of E-selectin ligands from SW620Mock and SW620FUT6 highlighted a similar band only in SW620FUT6 membrane proteins (Figure 6A). This staining pattern confirms the success of the isolation of L1CAM from the membrane protein of both cell lines and the presence of L1CAM among immunoprecipitated E-selectin ligands of SW620FUT6 and not SW620Mock membrane proteins.

Immunoprecipitated L1CAM and immunoprecipitated E-selectin ligands from SW620FUT6 membrane proteins were stained with HECA-452 (which recognizes sLe^X/A^ antigen) and E-Ig. The same band as in L1CAM staining appeared with sLe^X/A^ antigen and E-selectin ligands staining in both immunoprecipitated L1CAM and immunoprecipitated E-selectin ligands (Figure 6B). Together, these results showed that L1CAM was an E-selectin ligand in SW620FUT6 cells.

## 3. Discussion

Glycosylation alterations in CRC have been studied and reviewed by Holst and coworkers, who highlighted a strong increase of sLe^X/A^ during carcinogenesis [12]. These tetrasaccharides are the minimal binding determinants for the lectin family of E-selectin [32]. Engagement of circulating tumor cells (CTCs) to endothelial cell adhesion molecules is supported by E-selectin, essentially allowing tumor cells to firmly adhere to the endothelium and invade a new organ, therefore developing metastasis [8,9]. E-selectin is involved in the cancer metastasis formation process in CRC [17,18]. The sLe^X^ antigen expression has been highlighted in CRC and correlated with tumor metastasis and aggressiveness and with poorer prognosis and higher recurrence [33,34].

To further elucidate the role of E-selectin ligands in CRC, in this study we used the cell line SW620 derived from lymph node metastasis of colon cancer. Established cultured cell lines can present differences in term of glycosylation compared to observations made in human tissues [35], thus to better mimic tumor versus normal cells, we chose to use the fucosyltransferase 6 (*FUT6*) transfected SW620 cell line. FUT6 is upregulated in CRC and promotes the development of CRC via the PI3K/Akt signaling pathway [36]. Additionally, Trinchera et al. demonstrated that transfection of *FUT6*, coding α1,3-fucosyltransferase 6, resulted in sLe^X^ expression in different cell lines of gastrointestinal origin, including the SW620 cell line [29]. In agreement, in this study we observed that the SW620FUT6 cell line shows increased levels of sLe^X^ antigen and E-selectin ligands compared to SW620Mock as observed by two different staining techniques, flow cytometry and WB (Figure 1 and Figure 4). However, some discrepancies between the techniques have been shown. Indeed, in the Mock transfected cell line, the sLe^X/A^ and E-selectin ligands expression was null by WB while flow cytometry assays showed expression of both. The differences of expression observed between the flow cytometry and WB staining techniques have also been observed by our group when assessing the sLe^X/A^ and E-selectin ligands expression in immune cells [37]. Flow cytometry detects these antigens at cell surface glycoproteins and glycolipids, whereas WB detects E-selectin binding only among glycoproteins. Therefore, our results suggest that sLe^X/A^ decorated glycolipids may also be present at the surface of SW620Mock cells. As glycoprotein selectin ligands are more potent effectors of selectin binding than are glycolipids, it can be inferred that WB studies of E-selectin reactivity offer a better estimate of E-selectin binding capacity of cells under biological conditions [38].

Surprisingly, sLe^A^ antigen staining was found diminished by flow cytometry in *FUT6* transfected cell lines compared to the control. Since sLe^A^ and sLe^X^ biosynthesis require different chain types, and FUT6 can act only on type 2 chains (Figure 1A), substrate competition cannot be considered. Interestingly, *FUT3* mRNA expression was significantly reduced in SW620FUT6 cells, and FUT3 is the only enzyme with an α1,4 fucosyltransferase activity, essential for sLe^A^ biosynthesis. Thus, the reduction of sLeA observed in SW620FUT6 could be attributed to the lower expression of *FUT3*. However, both *FUT3* mRNA expression and sLe^A^ immunostaining were detected at extremely low levels for SW620Mock and FUT6 cell lines. These observations strengthen the conclusion that the strong increase in E-selectin ligand expression in SW620FUT6 cell line is directly linked to the sLe^X^ overexpression, without an influence from sLe^A^ antigen expression. *FUT5* expression coding FUT5, which is involved in the biosynthesis of sLe^x^ and other antigens as Le^X/Y^ antigens was also reduced in the SW620FUT6 cell line [39].

To deepen the characterization of SW620Mock and FUT6 cells, the *N*-glycan structures of the two cell lines membrane proteins from the two cell lines were determined by Liquid Chromatography Electrospray Ionization Tandem Mass Spectrometric (LC-ESI-MS/MS). CRC cell lines retain similar genetic profiles and functional characteristics to tumor tissues [40]. Moreover, Holst et al. and Chik et al. established by MS approaches *N*-glycan profiles for several CRC cell lines, which showed minor differences with CRC tumor tissues [35,41]. With regards to the SW620Mock cell *N*-glycan analysis, our results were consistent with the *N*-glycosylation profiling of the SW620WT cell line established by these two studies. Indeed, similar *N*-glycans such as high mannose type, hybrid and complex sialylated and/or fucosylated structures were highlighted in our analysis of SW620Mock *N*-glycans.

Our *N*-glycan profiles revealed typical pauci-/high-mannose structures identified in SW620Mock and SW620FUT6 cells, which are found elevated in other CRC cell lines and tissues [41,42,43,44,45]. Interestingly, we observed core-fucosylated pauci- and high-mannosidic type structures in both cell lines. Such structures were reported to be upregulated in CRC tumor tissues [44]. The presence of core-fucosylated paucimannosidic type structure is likely to be due to trimmed hybrid or complex structures by lysosomal exoglycosidases. Indeed, the increased activity of such enzymes (e.g., α-mannosidase, β-galactosidase and β-*N*-acetyl-hexosaminidase) has been reported in CRC and proposed as a potential CRC biomarker for diagnosis [46]. Nevertheless, only a few authors have reported the description of such structures in cancer and, of our knowledge, no functional role in cancer development has been studied for these. The presence of unusual core-fucosylated structures in CRC cell lines and tissues depicts perfectly how the glycosylation process can be disturbed in cancer cells and reinforces the importance of the field in the study of malignant tumor arise and development.

Other identified structures included hybrid type and mono- to penta-antennae branched complex type *N*-glycans. These structures are typically found elevated in CRC tissues especially sialylated or non sialylated Lewis type antigens [33,34,47,48]. As expected, more structures bearing sialylated Lewis type antigens (#34, #48, #49, #53, #59, #73, #74, #76, #77 and #78) were characterized in the SW620FUT6 cell line compared to the SW620Mock cell line. Thus, the *N*-glycan analysis showed for the first-time new structures carrying the sLe^X^ antigen in SW620 cells transfected with *FUT6*. The abundance of the antigen is in accordance with the different immunostaining techniques. A few other structures with different blood group and Lewis type antigens were distinguished between the SW620Mock and FUT6 cells such as one structure with type A antigen (#27) and two structures with Le^A/X^ antigen (#48 and #50) in SW620Mock cell *N*-glycans and three structures with Le^A/X^ antigen (#49, #57 and #70) and one structure with Le^B/Y^ antigen (#50). Nevertheless, these differences concern a small number of structures and are probably not related to FUT6 overexpression but to biological variation.

Interestingly, when comparing the *N*-glycan profiles of the two cell lines, SW620Mock cells presented seven additional bisected *N*-glycan structures against only two for SW620FUT6 cells. Furthermore, in SW620FUT6 cells, highly branched *N*-glycan structures with tetra- and penta-antenna were more abundant compared to *N*-glycans from SW620Mock cells. The presence of bisecting GlcNAc has been shown to reduce the formation of branched *N*-glycan structures [49] by preventing the action of branching GlcNAcTs, especially GlcNAcT-V. Moreover, reduced bisecting GlcNAc and increased branched *N*-glycan structures are typical glycosylation changes observed in CRC, correlating with tumor progression and metastasis formation. With the transfection of *FUT6*, SW620 cells acquire more aggressive and survival properties, with enhanced migration and immunomodulation abilities. In addition, more mono-sialylated *N*-glycans are observed in SW620FUT6 cells than Mock and an increase of sialylation is observed in CRC and associated with cancer progression, metastasis and poor prognosis [50,51].

Taken together, FUT6 overexpression in SW620 cells has a first direct, and expected, consequence of elevating the amount of sLe^X^ antigen, but seems also to have an indirect influence on the expression of other *N*-glycan structures associated with tumor progression. Consequently, manipulating the glycosylation in cell lines by overexpressing glycosyltransferase genes seems to have an unpredicted impact on the total glycan expression and requires deeper characterization of glycan structures.

We showed that SW620FUT6 cells possess improved migration ability compared to control transfected cells. This result proved that increased expression of sLe^X^ antigen and E-selectin ligands contributes to tumor cell migration. Similar observations were made in other cancer types where FUT6 overexpression resulted in E-selectin ligand expression and higher migration capability [52,53]. In a primary invasive ductal carcinoma cell line, we showed that inhibition of fucosylation inhibited sLe^X^ antigen and E-selectin ligand expression led to lower migration capacity and reduced IL-8 and TGF-β expression [54]. Furthermore, *FUT6* silencing reduced TGF-β-mediated EMT and inhibited migration in CRC cancer cells [55]. TGF-β is known for its immunosuppressive properties, and in fact the overexpression of antigens terminated by sialic acid leads to an immunosuppressive milieu due to differential recognition of the cancer cells by the immune system [56,57]. Additionally, TGF-β is involved in the modulation of FAKs, integrins and increases the capability of metastasis of CRC.

In summary, we found that FUT6 promotes cell migration, which is most likely to occur through increased expression of sLe^X^, inducing a profoundly altered phenotype in CRC cells. Therefore, as increased FUT6 expression is associated to similar roles in different cancer types, FUT6, sLe^X^ antigen and E-selectin ligands are potentially promising therapeutic targets for cancer treatment. It is possible that therapeutic approaches, such as those acting on the expression of other adhesion molecules and immunomodulators in CRC, [6,7,58] may also have implications in E-selectin ligand expression.

We demonstrated that the SW620FUT6 cell line expresses high levels of E-selectin ligands as shown by the use of E-selectin chimera in different staining techniques. We detected the E-selectin ligands on the cell surface and proteins presented high molecular weight (between 100 and 245 kDa). To identify the E-selectin ligands of SW620FUT6 cell lines, firstly we isolated those proteins and then used MS for identification. This allowed us to identify thirteen candidates, some of which were already known such as LAMP-2. Indeed, lysosomal membrane proteins, LAMP-1 and LAMP-2, have been identified in Colo-205 cell lines as E-selectin ligands and their expression levels mediated colon cancer cells adhesion to E-selectin [28,59]. Among the identified potential E-selectin ligands, we found integrins α6 and β1, which form the very late antigen 6 (VLA-6), the receptor for laminin involved in leukocyte binding under physiological shear flow condition [60]. Another identified E-selectin ligand candidate was PTPRJ. Different studies reported a role of the PTPRJ gene due to the loss of heterozygosity of this tumor suppressor gene early in colon neoplasia [61,62,63]. Nevertheless, there was no staining of PTPRJ in the SW620FUT6 cells by flow cytometry and, thus, PTPRJ was not further investigated.

In this study, we also demonstrated that the neural cell adhesion molecule L1 (L1CAM) is an E-selectin ligand in colon cancer cells. L1CAM was first identified in rats and was called NGF (nerve growth factor)-inducible large external glycoprotein [64]. Subsequently, the human L1CAM was identified, showing similarity to mouse, rat and chick homologs [65]. L1CAM is a type 1 membrane glycoprotein belonging to the immunoglobulin superfamily, composed of six immunoglobulin-like domains and five fibronectin type III domains with 21 potential *N*-glycosylation sites found on the extracellular portion. The protein is heavily glycosylated and migrates at ∼220 kDa in SDS-PAGE. Mainly found in the nervous system, L1CAM is involved in several processes such as neuronal migration, neurite fasciculation and synaptic plasticity, and its mutations cause severe neurological disorders [66,67,68]. Surprisingly, L1CAM expression has been highlighted in several types of cancer. Indeed, its expression correlates with tumor aggressiveness and metastasis in endometrial adenocarcinoma [69], breast cancer [70], non-small cell lung cancer [71] and melanoma [72]. It is also associated with reduced overall survival in esophageal squamous cell carcinoma [73], nerve invasion of pancreatic ductal adenocarcinoma [74] and chemoresistance in clear cell renal cell carcinoma [75] amongst others. In CRC patients, L1CAM expression has been associated with invasion, tumor progression, poor survival and metastasis [76,77,78]. Different mutations of L1CAM showed that the full-length protein enhance proliferation, cell motility and in vivo liver metastasis formation [79].

E-selectin ligand expression is directly involved in metastasis development as it has been shown in the Brodt et al. study, modulation of E-selectin expression in the liver promotes metastasis formation in vivo [80]. On the other hand, L1CAM expression in CRC is also linked to metastasis and by immunohistochemistry staining it has been shown that L1CAM expression is associated with the lymph node and bone marrow metastasis [78]. Furthermore, blocking L1CAM decreases adherence and migration of tumor cells from colon adenocarcinoma to the nervous system showing involvement of L1CAM in perineural invasion in CRC. [81]. In our study, we showed that an increase in sLe^X^ led to high E-selectin ligand expression and, curiously, resulted in increased L1CAM levels in SW620FUT6 cells. The role of L1CAM in CRC has been studied before but not as an E-selectin ligand. L1CAM is also expressed on leukocyte [82] and endothelial cells [83]. Although L1CAM can interact in homophilic binding, heterophilic binding with integrins has also been demonstrated [84]. Nevertheless, even if an interaction with E-selectin has never been demonstrated so far, our findings show that being an E-selectin ligand can also contribute to metastasis and the invasion role of L1CAM via improvement of endothelial cell interaction with CTCs. Another study reported a link between L1CAM expression and glycosylation, L1CAM overexpression improved cell migration in Chinese hamster ovary (CHO) cells through the increase of cell surface sialylation and fucosylation [85].

In conclusion, this work provided new insight on the L1CAM involvement in colon cancer metastasis mediated by glycan-specific interaction with E-selectin. The validation that L1CAM acts as a functional E-selectin ligand requires further studies. Namely in vitro studies with different cells models, in vivo studies to confirm role in metastasis and tissue analysis in cohorts of patients with CRC and other diseases. Nevertheless, the demonstration of a novel role for L1CAM as E-selectin ligand, increases the interest to consider this glycoprotein as a prognostic marker or a future therapeutic target in CRC.

## 4. Materials and Methods

### 4.1. Reagents and Antibodies

PBS pH 7.4: 1.47 mM KH_2_PO_4_, 4.29 mM Na_2_HPO_4_, 137 mM NaCl and 2.68 mM KCl, pH adjusted to 7.4 with 400 mM NaOH solution, diluted in sterile Milli-Q^®^ H_2_O to 1 L.

For immunoprecipitation, wash buffer: 150 mM NaCl, 2 mM CaCl_2_, 50 mM Tris pH 7.4, 1 Complete, Mini, EDTA-free Protease Inhibitor Cocktail Tablet (Roche, Indianapolis, IN, USA), 2% NP-40; blocking buffer: wash buffer, 1% bovine serum albumin (BSA); denaturing buffer: wash buffer without NP-40, 0.5% SDS and 40 mM DTT.

For WB, TBS pH 7.6: 15.4 mM Tris-HCl and 137 mM NaCl, pH adjusted to 7.6 with 400 mM NaOH solution, diluted in sterile Milli-Q^®^ H_2_O to 1 L; wash buffer: TBS-0.1% Tween 20 and blocking buffer: wash buffer 10% dry milk.

Recombinant Human E-Selectin/CD62E Fc Chimera Protein (E-selectin-Ig chimera, here called E-Ig) was purchased from R&D Systems (Minneapolis, MS, USA). Rat anti-human cutaneous lymphocyte antigen (CLA) mAb, clone HECA-452, was from BioLegend (San Diego, CA, USA). Mouse anti-human CA19-9 mAb, clone CA19-9-203, was from Abcam (Cambridge, UK). Rat anti-mouse CD62E mAb, mouse anti-human CD15s mAb, clone CSLEX1, goat anti-mouse immunoglobulins (Ig) conjugated with horseradish peroxidase (HRP) and goat anti-rat Ig conjugated with allophycocyanin (APC) were from BD Biosciences (San Jose, CA, USA). Goat anti-mouse Ig FITC was from Dako (Santa Clara, CA, USA). Goat anti-rat IgG (H+L) HRP, goat anti-rat IgM HRP and goat anti-mouse IgG FITC were from Southern Biotech (Birmingham, AL, USA). Anti-human IgG (Fc Specific)-fluorescein isothiocyanate (FITC) was from Sigma-Aldrich (St. Louis, MO, USA). Mouse anti-human L1CAM mAb, clone UJ127.11, mouse anti-human PTPRJ mAb, clone F-12, mouse anti-human integrin α6 mAb, clone BQ16, mouse anti-human integrin β1 mAb, clone JB1B and mouse anti-human β-tubulin mAb, clone D-10, were from Santa Cruz Biotechnology (Dallas, TX, USA).

### 4.2. Cell Culture

The human colon cancer cell lines, *FUT6*-transfected SW620 (SW620FUT6) and Mock control (SW620Mock), were obtained as described [29] and were grown in sterile condition at 37 °C without CO_2_ in Leibovitz L-15 medium, supplemented with 10% (*v*/*v*) fetal bovine serum, 2 mM L-glutamine, 100 U·mL^−1^ penicillin and 100 µg·mL^−1^ streptomycin. Cultures were passaged using 1X trypsin-EDTA.

### 4.3. Flow Cytometry

Cell surface expression of sLe^X^, sLe^A^, sLe^X/A^ and E-selectin ligand were analyzed by flow cytometry as previously described [25]. L1CAM, PTPRJ, integrin α6 and integrin β1 staining was performed 30 min at 4 °C with primary mAb followed by incubation with the fluorescent-labeled secondary antibody. Flow cytometry was performed using an Attune Acoustic Focusing Cytometer (Applied Biosystems, Waltham, MA, USA). To analyze the results and for the figures, FlowJo software was used. Median fluorescent intensity (MFI) ratios were determined by the MFI values of stained *FUT6* transfected cell lines divided by MFI values of stained Mock transfected cell lines values and used for statistical analysis for E-Ig, HECA-452, CD15s and CA19-9 staining. MFI values were used for statistical analysis for L1CAM, PTPRJ, integrin α6 and integrin β1 staining.

### 4.4. Membrane Proteins Extraction

Membrane proteins from SW620Mock and SW620FUT6 cells were obtained using Mem-PER Plus Membrane Protein Extraction Kit (Thermo Scientific, Waltham, MA, USA) following the manufacturer’s instruction. Protease inhibitor (Roche) was added to extracted membrane and membrane-associated proteins. Proteins concentrations were evaluated using Pierce BCA Protein Assay Kit (Thermo Scientific).

### 4.5. Immunoprecipitation

For each assay, 100 µg of membrane extracted proteins was cleared with recombinant Protein (rProtein) G Agarose (Invitrogen™, Thermo Scientific), incubated at 4 °C for two hours with agitation. In parallel, rProtein G-agarose beads were incubated with 3 or 2 µg of antibody (E-Ig, in the presence of 2 mM Ca^2+^, or mAb mouse anti-human L1CAM, respectively) overnight at 4 °C with shaking. Cleared membrane proteins were added to the rProtein G-agarose with antibody for six to eight hours at 4 °C with agitation. After centrifugation, the whole cleared membrane proteins from immunoprecipitates were stored for Western blot analysis and the immunoprecipitated (IP) proteins were obtained by boiling the rProtein G-agarose with denaturing buffer for ten minutes. Supernatant containing IP proteins were conserved at −80 °C.

### 4.6. SDS-PAGE and Western Blot

Cells were lysed in wash buffer for immunoprecipitation (see Reagents). Cell lysates were agitated using a vortex overnight at 4 °C, and supernatants were cleared by centrifugation for 10 min at 10,000× *g* and stored at −80 °C until use. Protein concentrations were evaluated using a Pierce BCA Protein Assay Kit (Thermo Scientific). Of proteins 20 µg obtained from total cell lysates, 10 µg of membrane extracted proteins, 20 µL of cleared proteins from immunoprecipitation or 10 µL of IP proteins electrophoresed through 8 or 6% (*v*/*v*) SDS-PAGE. Prestained protein ladder (Abcam) was included in adjacent lanes in all experiments. Resolved proteins were transferred to polyvinylidene difluoride (PVDF) membranes and blocked for one hour at room temperature with blocking buffer. Membranes were incubated with primary mAbs overnight at 4 °C under agitation, for E-Ig staining all buffers were supplemented with 2 mM CaCl_2_. For E-Ig staining only, incubation with anti-CD62E mAb for an hour at room temperature under agitation were done. Following this, membranes were incubated with appropriate HRP-conjugated secondary antibodies at room temperature for one hour under agitation. Immunoblots were visualized by using by chemiluminescence detection Lumi-light (Roche) β-tubulin protein expression level was analyzed as a loading control.

### 4.7. Mass Spectrometry

#### 4.7.1. E-selectin Ligand Identification

##### Mass Spectrometry

IP proteins with E-Ig were subjected to clean up by a short SDS-PAGE run (NuPAGE™ 4–12% Bis-Tris Protein Gels, Invitrogen™) and stained with SimplyBlue™ Safe Stain (Invitrogen™) for three hours followed by washing with distilled water. Bands corresponding to the whole lane were cut and proteins were in-gel digested with trypsin after reduction (dithiothreitol 10 mM) and alkylation (iodoacetamide 50 mM) using a Proteineer DP digestion robot (Bruker Daltonics). Tryptic peptides were then extracted from the gel slices, lyophilized, dissolved in solvent A (water/0.1 formic acid (FA) *v*/*v*) and subsequently analyzed by online C18 nano-HPLC MS/MS with a system consisting of an Easy nLC 1200 gradient HPLC system (Thermo, Bremen, Germany) and a LUMOS mass spectrometer (Thermo). Fractions were injected onto a homemade precolumn (100 μm × 15 mm; Reprosil-Pur C18-AQ 3 μm, Dr. Maisch, Ammerbuch, Germany) and eluted via a homemade analytical nano-HPLC column (20 cm × 75 μm; Reprosil-Pur C18-AQ 3 μm) using a gradient from 10 to 40% solvent B (20/80/0.1 water/acetonitrile/FA *v*/*v*/*v*) in 20 min. The nano-HPLC column was drawn to a tip of ∼5 μm and acted as the electrospray needle of the MS source. The LUMOS mass spectrometer was operated in data-dependent MS/MS mode (cycle time 3 s) with normalized collision energy of 32% and recording of the MS2 spectrum in the Orbitrap. In the master scan (MS1) the resolution was 120,000, and the scan range was from *m*/*z* 400–1500 at an AGC target of 400,000 with maximum fill time of 50 ms. Dynamic exclusion after *n* = 1 with an exclusion duration of 10 s was applied. Charge states 2–5 were included for MS2. For this, precursors were isolated with the quadrupole with an isolation width of 1.2 Da. The MS2 scan resolution was 30000 with an AGC target of 50,000 with a maximum fill time of 60 ms.

##### Data Analysis

MS/MS data were searched against a human protein database (UniProt, 67915 entries) using the Mascot search algorithm (Matrix Science, London, UK; version 2.2.04). Trypsin was selected as the enzyme (up to two missed cleavages were allowed) and the MS and MS/MS tolerance were 10 p.p.m. and 0.02 Da, respectively. Carbamidomethylation of cysteine was set as a fixed modification and oxidation of methionine was specified as a variable modification. Scaffold (version Scaffold_4.8.4, Proteome Software Inc., Portland, OR) was used to validate MS/MS based peptide and protein identifications applying an FDR (false discovery rate) of 1% at minimum two unique peptides identified with a 95% peptide threshold identification.

#### 4.7.2. Glycoanalysis

##### *N*-glycan Release

Membrane proteins extracted from SW620Mock and SW620FUT6 cells were dried down. Reagents used for *N*-glycan released were obtained from the LudgerZyme™ PNGase F Release Kit (LZ-rPNGaseF-kit, Ludger, UK). Samples were resuspended in water and denatured at 100 °C for 10 min with denaturation buffer (5% SDS 400 mM DTT). Following the denaturation, after the samples were allowed to cool to room temperature, samples were incubated with PNGase F in reaction buffer (500 mM sodium phosphate, pH 7.5 at 1× dilution) and NP−40 at 37 °C overnight. Released *N*-glycans were dried with vacuum centrifugation and then converted to aldoses by incubating with 1% formic acid solution at room temperature for 45 min. Remaining proteins and enzymes were removed using a LudgerClean™ Protein Binding Membrane Plate (LC-PBM-96, Ludger, Oxford, UK). Finally, eluted *N*-glycans were dried using vacuum centrifugation prior to labeling.

##### Glycan Labeling

Glycans were labeled with procainamide using a LudgerTag™ Procainamide Glycan Labeling Kit with sodium cyanoborohydride (LT-KPROC-24, Ludger) [86]. The procainamide dye solution (DMSO, acetic acid, procainamide, sodium cyanoborohydride and water) was prepared following the manufacturer’s protocol and added to the samples. Labeling was done at 65 °C for one hour and labeled samples were then cleaned-up.

##### *N*-glycan Clean-Up

Clean up of samples and removal of excess dye was performed using a Ludger Clean plate (LC-PROC-96, Ludger) following the manufacturer’s protocol. Briefly, samples were added to the plate in acetonitrile, washed 3 times with acetonitrile (200 µL) and eluted in water (2 × 100 µL). Purified labeled *N*-glycans were stored at 4 °C until they could be processed. For longer term storage −20 °C was used.

##### HILIC-UHPLC Analysis on a Dionex Ultimate 3000 with Inline MS

Samples and standards were analyzed by liquid chromatography electrospray ionization tandem mass spectrometry (LC-ESI-MS/MS). Procainamide labeled glycans were dried using vacuum centrifugation. Samples were resuspended in pure water. Samples were injected in 25% aqueous/75% acetonitrile; injection volume 25 µL. Samples made up as follows: IP, 12.5 µL plus 3.57 µL acetonitrile; mAb, 12.5 µL plus 37.5 µL acetonitrile; membrane proteins, 12.5 µL plus 37.5 µL acetonitrile; standards and 25 µL plus 75 µL acetonitrile.

Samples were analyzed by HILIC-LC on an Ultimate 3000 UHPLC using a BEH-Glycan 1.7 µm, 2.1 × 150 mm column (Waters) at 40 °C with a fluorescence detector (λex = 310 nm, λem = 370 nm), controlled by Bruker HyStar 3.2 and Chromeleon data software version 7.2. Buffer A was 50 mM ammonium formate made from LudgerSep N Buffer stock solution, pH 4.4 (LS-N-BUFFX40) and buffer B was acetonitrile (acetonitrile 190 far UV/gradient quality; Romil #H049). Gradient conditions were: 0–53.5 min, 24–49.0% A (0.4 mL·min^−1^); 53.5–55.5 min, 49.0–0% A (0.4–0.2 mL·min^−1^); 55.5–57.5 min, 100% A (0.2 mL·min^−1^); 57.5–59.5 min, 100–24% A (0.2 mL·min^−1^); 59.5–65.5 min, 24% A (0.2 mL·min^−1^); 65.5–66.5 min 24% A (0.2–0.4 mL·min^−1^) and 66.5–70 min, 24% A (0.4 mL·min^−1^). Sensitivity S5 was used for the standards and membrane protein samples. Chromeleon data software version 7.2 with a cubic spline fit was used to allocate glucose unit (GU) values to peaks. Procainamide labeled glucose homopolymer was used as a system suitability standard and an external calibration standard for GU allocation for the system [87].

Analysis was performed using a Bruker amaZon Speed ETD electrospray mass spectrometer, which was coupled directly after the UHPLC FD without splitting. The instrument scanned samples in the maximum resolution mode, positive ion setting, MS scan + three MS/MS scans, nebulizer pressure 14.5 psi, nitrogen flow 10 L·min^−1^ and capillary voltage 4500 Volts. MS/MS was performed on three ions in each scan sweep with a mixing time of 40 ms. Mass spectrometry data were analyzed using the Bruker Compass DataAnalysis 4.1 software. LC-ESI-MS/MS chromatogram analysis was performed using Bruker Compass DataAnalysis 4.4 and GlycoWorkbench software. Structures were identified by comparing LC, MS and MS/MS data.

### 4.8. RT-qPCR

RNA extraction was made using GenElute™ Mammalian Total RNA Miniprep Kit (Sigma-Aldrich) following the manufacturer’s instruction. On-Column DNase I Digestion Set (Sigma-Aldrich) was used to eliminate genomic DNA from RNA extracts. RNA was converted to cDNA using High-Capacity cDNA Reverse Transcription Kit (Applied Biosystems) following the manufacturer’s instruction. qPCR assays were performed in the Rotor-Gene 6000 system (Corbett Research) using TaqMan™ Fast Universal PCR Master Mix (2X), no AmpErase™ UNG (Applied Biosystems), TaqMan™ Gene Expression Assay primers and probe with the following ID: Hs00356857_m1 (*FUT3*), Hs01106466_s1 (*FUT4*), Hs00704908_s1 (*FUT5*), Hs03026676_s1 (*FUT6*), Hs00237083_m1 (*FUT7*), Hs00276003_m1 (*FUT9*), Hs00327091_m1 (*FUT10*) and Hs00543033_m1 (*FUT11*). The relative mRNA levels were normalized against the mean of the *β-actin* (Hs99999903_m1) and *GAPDH* (Hs99999905_m1) expression and calculated by adapted formula 2^−ΔCt^ × 1000, which infers the number of mRNA molecules of the gene of interest per 1000 molecules of the endogenous controls [88]. ΔCt stands for the cycle threshold difference between the target gene and the endogenous control genes. Cycle thresholds were obtained by analyzing the qPCR results with the Rotor Gene 6000 software.

### 4.9. Wound Healing Assay

The Ibidi Culture-Inserts 2 Wells were used according to the manufacturer’s instructions to perform the wound healing assay. Culture inserts were placed in each well, and a cell suspension concentration of 7.5 × 10^5^ cells.mL^−1^ of both cell lines were seeded to obtain confluent layers after 24 h. The inserts were then removed, and the gap formed in the cell layer was monitored for seven days. To monitor the migration capability, microscope with a digital camera was used to register pictures of the wound each day. For quantitative measurements, the pictures were analyzed using ImageJ software to measure the area of the wound free of cells.

### 4.10. Statistical Analysis

Statistical analysis was performed using GraphPad Prism 6 (GraphPad Software, Inc., San Diego, CA, USA). Data were analyzed using D’Agostino and Pearson omnibus normality test to determine normal distribution, then unpaired Student’s *t* test for parametric data or Mann–Whitney test for non-parametric data were applied. Differences were considered statistically significant when *p* < 0.05 (*), *p* < 0.01 (**), *p* < 0.005 (***) and *p* < 0.0001 (****).

## Figures and Tables

**Figure 1 ijms-21-08286-f001:**
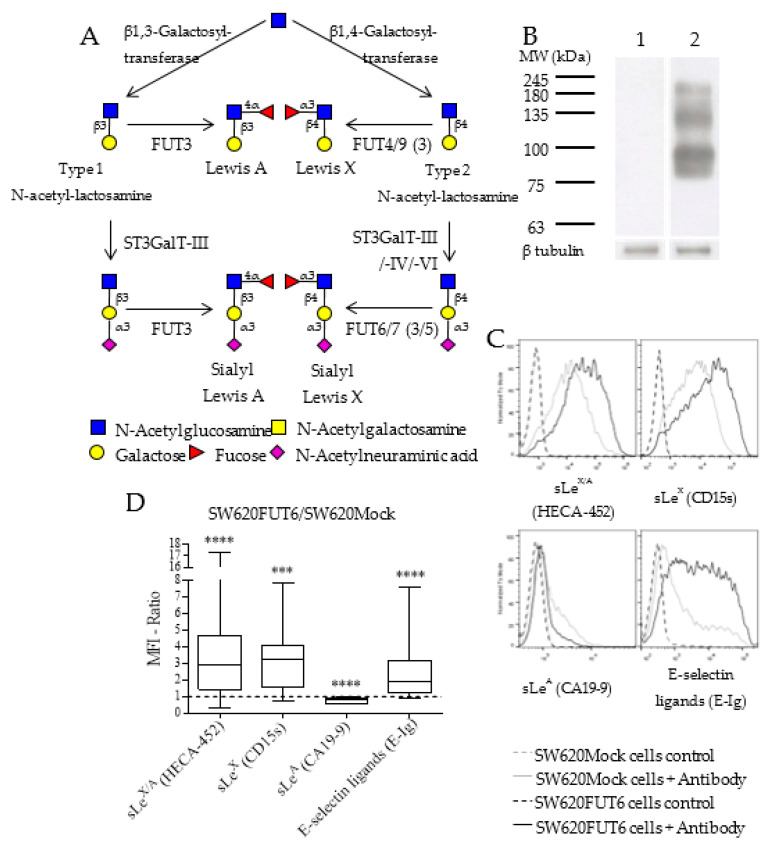
Comparison of sialyl Lewis X/A and E-selectin ligand expression in SW620Mock and SW620FUT6 cell lines. (**A**) Biosynthesis of Le^A^, Le^X^ and their sialylated version involves multiple enzymes such as galactosyltransferases, β galactoside α2,3 sialyltransferases (ST3GalTs) and FUTs. Le^A^ and sLe^A^ are formed from type 1 *N*-acetyllactosamine (LacNAc) structure, i.e., Galβ1,3GlcNAc, while Le^X^ and sLe^X^ antigens are formed from type 2 LacNAc structure, i.e., Galβ1,4GlcNAc. (**B**) Whole cell lysates were resolved by sodium dodecyl sulfate polyacrylamide gel electrophoresis (SDS-PAGE) and immunoblotted. Protein extracted from SW620Mock (lane 1) and SW620FUT6 cells (lane 2) were stained with rat anti-human HECA-452 mAb followed by goat anti-rat IgM HRP by Western blot (WB). β-tubulin protein expression level was analyzed as loading control. (**C**) SW620Mock cells (grey) and SW620FUT6 transfected cells (black) were stained with primary antibody (specified below each histogram) followed by fluorescent secondary antibody in PBS, supplemented with Ca^2+^ for E-Ig staining. As a negative control, cells were stained with secondary antibody only (dashed lane), or in PBS with EDTA for E-Ig staining. (**D**) SW620FUT6/SW620Mock MFI ratios were determined for each independent staining experiment; for sLe^X/A^ (HECA-452) staining *n* = 17 *p* < 0.0001 (****); for sLe^X^ (CD15s) staining *n* = 17 *p* = 0.0001 (***); for sLe^A^ (CA19-9) staining *n* = 12 *p* < 0.0001 (****) and for E-selectin ligands (E-Ig) staining *n* = 14 *p* < 0.0001 (****).

**Figure 2 ijms-21-08286-f002:**
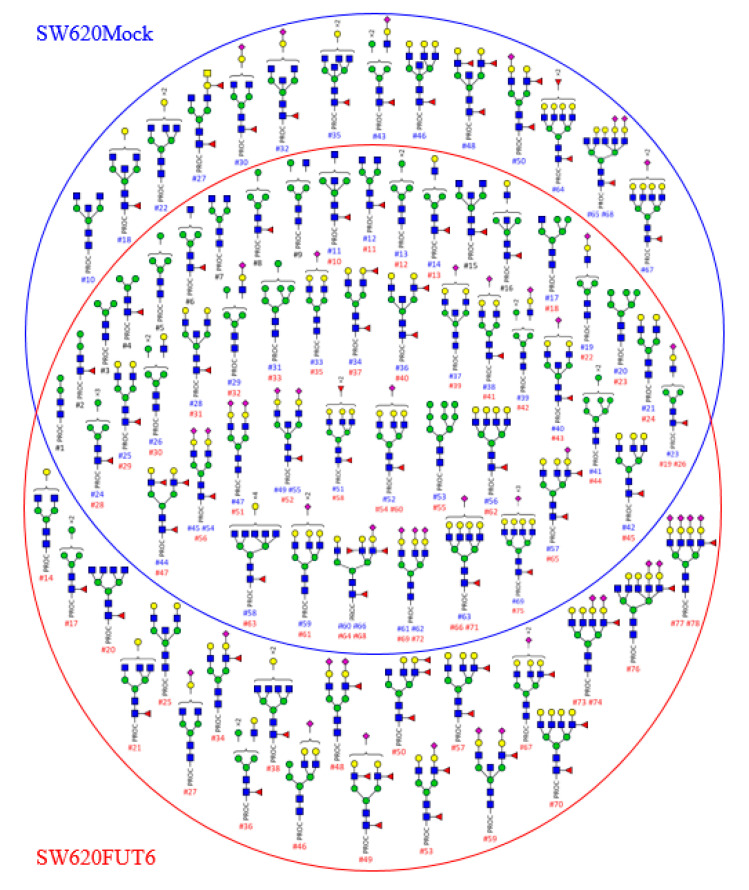
*N*-glycans identified by HILIC-UHPLC-MS from membrane proteins of SW620Mock and FUT6 cells. Schematic representation of the *N*-glycans released from SW620Mock cells (inside the blue circle) and from SW620FUT6 cells (inside the red circle) membrane proteins. Structures identified in both cell lines are shown where the two circles overlap. Sixty-nine *N*-glycan structures were identified for SW620Mock membrane proteins and 78 for SW620FUT6 membrane proteins. Y- and B-ion fragments identification, which allowed *N*-glycan composition determination, is depicted in Appendix A for SW620Mock and SW620FUT6 cells membrane proteins, respectively. In the Appendix A, a peak ID has been attributed to each structure, which corresponds to the #number (#) under each *N*-glycan (blue for *N*-glycans from SW620Mock proteins, red for *N*-glycans from SW620FUT6 proteins and black for identical peak ID number attributed to *N*-glycans from both SW620Mock and FUT6 proteins). Structures are depicted with the following notation: PROC: procainamide; blue square: *N*-acetylglucosamine; green circle: Mannose; yellow circle: Galactose; red triangle: Fucose; purple diamond: *N*-acetylneuraminic acid.

**Figure 3 ijms-21-08286-f003:**
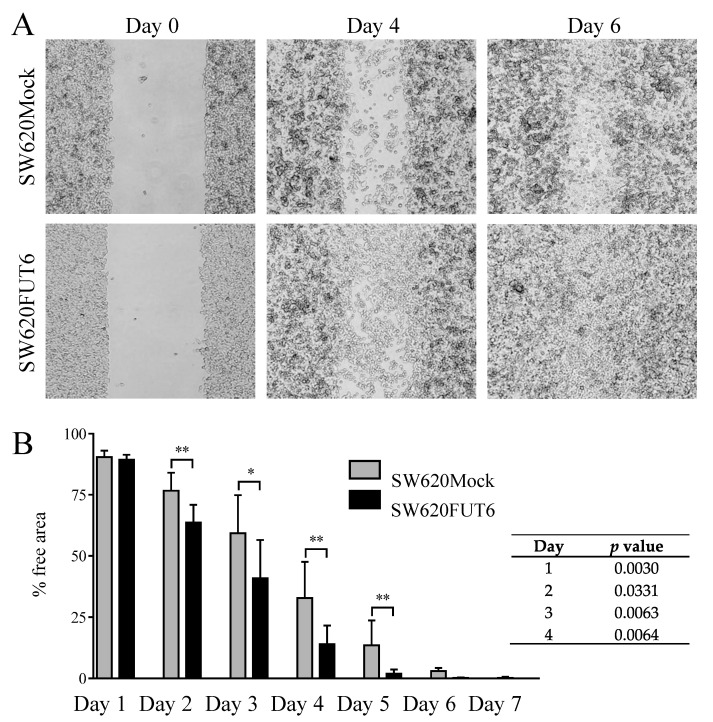
FUT6 overexpression in SW620 cells led to increased cell migration and wound healing. (**A**) Pictures of one of the scratch wound healing assays, performed using culture inserts, from day 0 (left), day 4 (middle) and day 6 (right) with SW620Mock cells (upper) and SW620FUT6 cells (lower) obtained with an inverted microscope. (**B**) Means with standard deviation of percentage of free areas for 8 independent experiments are represented, free areas were determined from pictures using ImageJ and converted into percentage values. At day 2, 3, 4 and 5, statistical analysis using an unpaired Student’s *t*-test demonstrated significant higher migration ability for SW620FUT6 cells compared to SW620Mock cells. Differences were considered statistically significant when *p* < 0.05 (*), and *p* < 0.01 (**).

**Figure 4 ijms-21-08286-f004:**
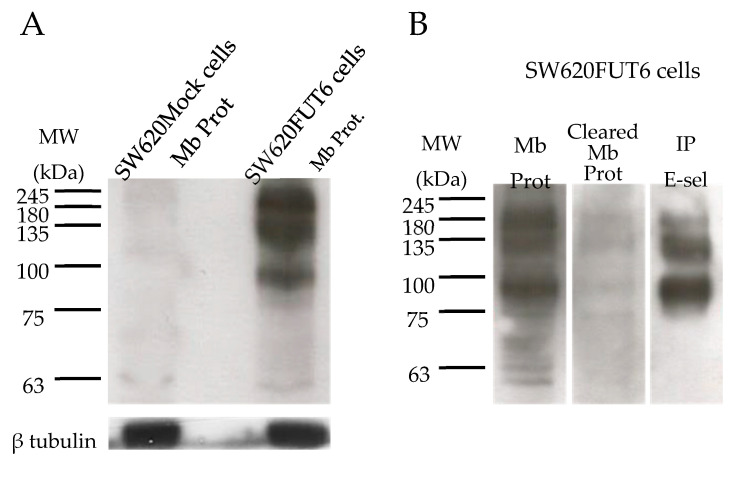
SW620FUT6 cells express E-selectin ligands. (**A**) Membrane proteins (Mb Prot) from SW620Mock and SW620FUT6 cells were stained with E-Ig plus anti-mouse CD62E plus anti-rat IgG (H+L) HRP in PBS with Ca²^+^ by WB. β-tubulin protein expression level was analyzed as loading control. (**B**) Membrane proteins, cleared membrane proteins from immunoprecipitated (IP) E-selectin ligands and IP E-selectin ligands (IP E-sel) from 100 µg. of SW620FUT6 cell membranes were stained with rat anti-human HECA-452 mAb followed by goat anti-rat IgM HRP by WB.

**Figure 5 ijms-21-08286-f005:**
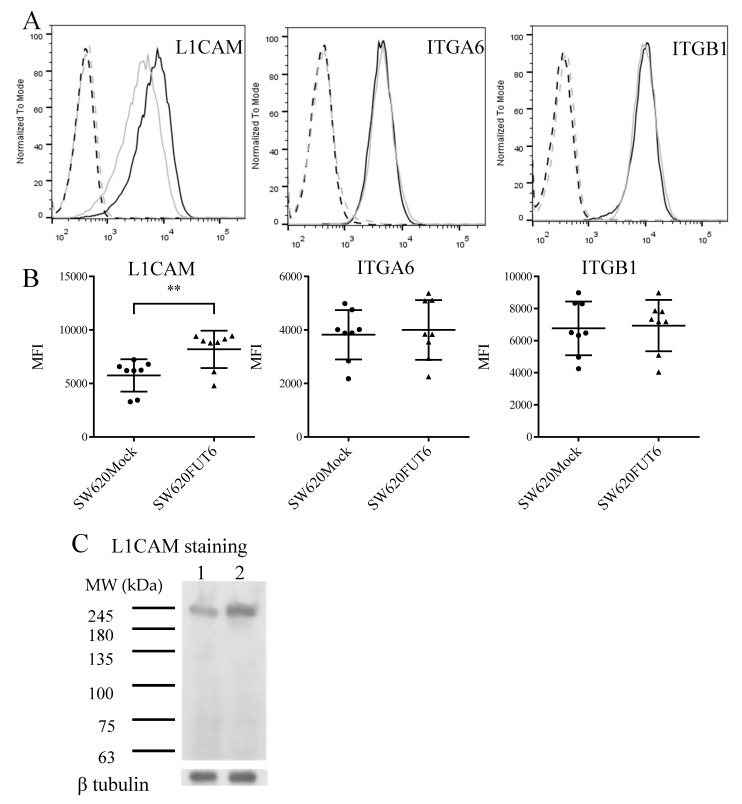
E-selectin ligands expression in SW620 cell lines. (**A**) Flow cytometric analysis of L1CAM, ITGA6 and ITGB1 expression. SW620Mock (grey filled lane) and SW620FUT6 (black filled lane) cells were stained with corresponding mAb plus anti-mouse IgG FITC and analyzed by flow cytometry. Negative controls for SW620Mock and SW620FUT6 (grey and dark dashed lane, respectively) were stained with secondary antibody only (anti-mouse IgG FITC). (**B**) Median fluorescent intensity (MFI) values (filled circles and triangles) of 8 biological independent experiments of anti-L1CAM, anti-ITGA6 and anti-ITGB1 mAb plus anti-mouse IgG FITC staining obtained by flow cytometry. The data are represented in scatter plot with mean ± standard deviation (SD). L1CAM expression was significantly higher in SW620FUT6 cells compared to SW620Mock cells, ** *p* = 0.0096 (unpaired Student’s *t*-test). (**C**) Membrane proteins from SW620Mock (lane 1) and SW620FUT6 cells (lane 2) were stained with anti-L1CAM mAb plus anti-mouse Ig HRP in PBS by WB. β-tubulin protein expression level was analyzed as loading control.

**Figure 6 ijms-21-08286-f006:**
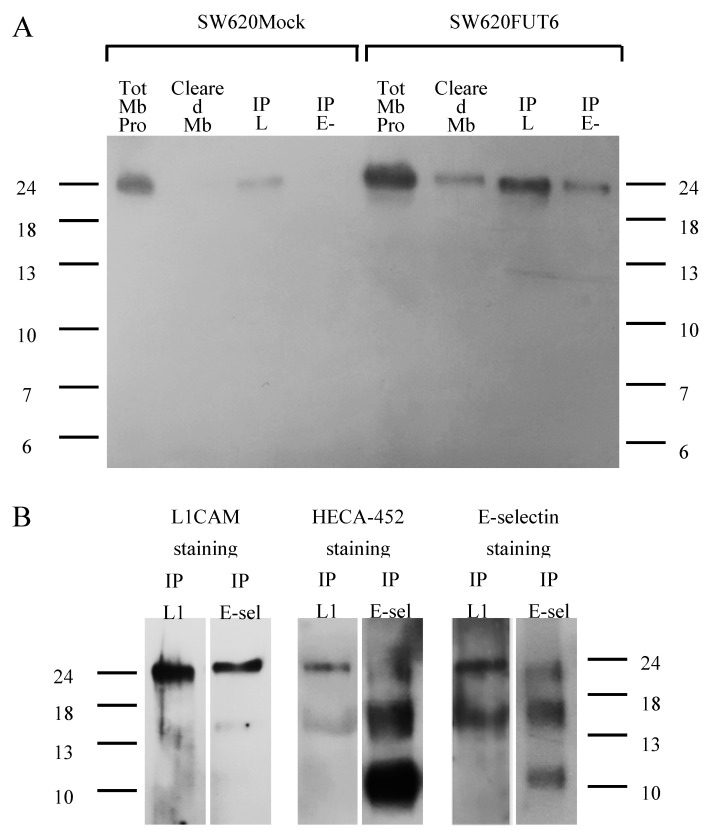
Identification of L1CAM as an E-selectin ligand in SW620FUT6 cells. (**A**) L1CAM and E-selectin ligands immunoprecipitations from 100 µg of SW620Mock and FUT6 cells membrane proteins, and cleared membrane proteins from immunoprecipitated (IP) L1CAM, were stained with anti-L1CAM mAb plus anti-mouse Ig HRP in PBS by WB. IP E-selectin ligands from SW620FUT6 cells showed a stained band corresponding to L1CAM protein suggesting that L1CAM is an E-selectin ligand, in contrast to IP E-selectin ligands from SW620Mock cells. (**B**) IP L1CAM and IP E-selectin ligands from 100 µg of SW620FUT6 membrane proteins were stained with anti-human L1CAM mAb plus anti-mouse Ig HRP in PBS (left), with HECA-452 (anti-human CLA mAb) plus anti-rat IgM HRP in PBS (middle) and with E-Ig plus anti-mouse CD62E plus anti-rat IgG (H+L) HRP in PBS with Ca²^+^ (right) by WB. Abbreviation: Tot: total, Mb Prot: membrane proteins, IP: immunoprecipitated, L1: L1CAM, E-sel: E-selectin ligands.

**Table 1 ijms-21-08286-t001:** α1,3/4 fucosyltransferases (*FUTs*) gene expression in SW620FUT6 and Mock cell lines. Values represent relative number of mRNA molecules obtained by RT-qPCR and correspond to the mean ± SEM (standard error of mean) of the gene of interest per 1000 molecules of the endogenous controls (see “Materials and methods” RT-qPCR section), *p* value obtained with Mann–Whitney test, n = 5.

	*FUT3*	*FUT4*	*FUT5*	*FUT6*	*FUT7*	*FUT10*	*FUT11*
SW620Mock	0.001 ± 0.0001	12.6 ± 1.1	0.01 ± 0.004	2.7 ± 0.8	0.002 ± 0.0002	5.1 ± 0.8	1.5 ± 0.4
SW620FUT6	0.0002 ± 0.00005	14.3 ± 4.0	0.002 ± 0.0008	107.3 ± 40.0	0.002 ± 0.0005	4.3 ± 0.8	1.4 ± 0.3
*p* value	** *p* = 0.0079	N.S.	* *p* = 0.0317	** *p* = 0.0079	N.S.	N.S.	N.S.

Abbreviations: *FUT*, fucosyltransferase; N.S., not significant. Differences were considered statistically significant when *p* < 0.05 (*), and *p* < 0.01 (**).

**Table 2 ijms-21-08286-t002:** List of E-selectin ligands in SW620FUT6 cells identified by mass spectrometry†. † From SW620Mock and SW620FUT6 cells extracted membrane proteins, four immunoprecipitations with E-Ig were performed and analyzed by LC–MS/MS. The present list shows the glycosylated‡ immunoprecipitated E-selectin ligands only identified in SW620FUT6 cells (Appendix A for glycosylated‡ IP E-selectin ligands identified in both cell lines). § Proteins identified are described with the number of unique peptides for each experiment and the sum of the total spectrum count from the four experiments. ‡ Protein information on glycosylation status and subcellular location were extracted from the UniProtKB database.

Protein Name	Gene Name	MW (kDa)	UniProtKB Entry	Exclusive Unique Peptide Counts §				Total Spectrum Count (sum) §	Subcellular Location ‡
				Exp. 1	Exp. 2	Exp. 3	Exp. 4		
Neural cell adhesion molecule L1	L1CAM	140	P32004	31	20	30	27	118	Plasma membrane
Integrin α6	ITGA6	127	P23229	11	31	12	10	91	Plasma membrane
Receptor-type tyrosine-protein phosphatase eta	PTPRJ	146	Q12913	21	9	18	11	58	Plasma membrane
Integrin β1	ITGB1	88	P05556	6	11	7	5	39	Plasma membrane, recycling endosome
Cation-independent mannose-6-phosphate receptor (Insulin-like growth factor 2 receptor)	IGF2R	274	P11717	16	3	16	7	39	Lysosome membrane
Receptor-type tyrosine-protein phosphatase alpha	PTPRA	91	P18433	10	3	4	5	22	Membrane
Leucyl-cystinyl aminopeptidase	LNPEP	117	Q9UIQ6	7	9	6	5	22	Plasma membrane, secreted
Carboxypeptidase D	CPD	153	O75976	2	0	6	6	17	Plasma membrane
Lysosome-associated membrane glycoprotein 2	LAMP2	45	P13473	3	3	3	3	12	Lysosome/endosome/plasma membrane
CD109 antigen	CD109	162	Q6YHK3	3	2	2	4	12	Plasma membrane
Golgi membrane protein 1	GOLM1	45	Q8NBJ4	2	2	2	4	10	Golgi apparatus membrane
Plexin-D1	PLXND1	212	Q9Y4D7	2	0	1	4	7	Plasma membrane
Zymogen granule protein 16 homolog B	ZG16B	23	Q96DA0	0	2	0	2	4	Secreted

Abbreviation: MW, Molecular Weight; Exp., Experiment.

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
