# Peer review of "L1CAM as an E-selectin Ligand in Colon Cancer"

_ijms, 2020, doi:10.3390/ijms21218286_

Round 1
Reviewer 1 Report
Title: “L1CAM as an E-selectin ligand in colon cancer”
Authors: Fanny M. Deschepper, Roberta Zoppi, Martina Pirro, Paul J. Hensbergen, Fabio Dall’Olio, Maximillianos Kotsias, Richard A. Gardner, Daniel I.R. Spencer, Paula A. Videira
Summary:
Metastasis is the main cause of death among colorectal cancer (CRC) patients.
Deschepper and co-authors have characterized the glycoengineered cell line SW620 transfected with the fucosyltransferase 6 (FUT6) coding for the α1,3-fucosyltransferase 6 (FUT6) which is the main enzyme responsible for the synthesis of sLeX in CRC. The SW620FUT6 cell line expressed high levels of sLeX antigen and E-selectin ligands. Moreover, it displayed increased migration ability. E-selectin ligand glycoproteins were isolated from the SW620FUT6 cell line, identified by mass spectrometry, and validated by flow cytometry and Western blot (WB). The most prominent E-selectin ligand we identified was the neural cell adhesion molecule L1 (L1CAM). The association of L1CAM with metastasis in cancer and the novel role as E-selectin counter-receptor contributes to understanding the molecular mechanism involving L1CAM in metastasis formation.
Comments:
A well-written article highlighting that the E-selectin and its carbohydrate ligands, including sialyl Lewis X (sLeX) antigen, are key players in the binding of circulating tumor cells to the endothelium, which is one of the major events leading to organ invasion.
Minor Points:
- With regard to migration, EMT, FAK, Integrins and NF-kB should be described in more detail in the introduction and discussed in the discussion.
Please add additional reference:
Buhrmann et al, 2014, PLOS ONE, Volume 9, e107514.
Buhrmann et al., 2017, Nutrients 9, 1073;
Buhrmann et al., 2020, Biomedicines 8, 236;
- Please write in the whole paper instead Integrin beta-1 or β-1: Integrin β1, α-6: α6 etc.
- In Figure 4B: control protein is missing.
- In Figure 6A-B: control protein is missing.
- The authors should discuss the limitations of the study.
- Whole blot of raw data of western blot must be shown.
Author Response
Thank you for reviewing the manuscript entitled L1CAM as an E-selectin ligand in colon cancer by Deschepper et al that has been submitted for consideration in the International Journal of Molecular Sciences
We have carefully considered the comments of the peer reviewers and have revised the manuscript accordingly. The changes performed in the revised version can be seen as highlighted text. We have also prepared a detailed response to each of the reviewers’ comments, which are written below after the reviewers’ original text.
We are honestly grateful to you, whose careful reading of the manuscript, and suggestions have helped us to improve the manuscript.
We sincerely expect that you find this new version of the manuscript now acceptable.
# Reviewer 1 Comments:
A well-written article highlighting that the E-selectin and its carbohydrate ligands, including sialyl Lewis X (sLeX) antigen, are key players in the binding of circulating tumor cells to the endothelium, which is one of the major events leading to organ invasion. With regard to migration, EMT, FAK, Integrins and NF-kB should be described in more detail in the introduction and discussed in the discussion.
Please add additional reference: Buhrmann et al, 2014, PLOS ONE, Volume 9, e107514. Buhrmann et al., 2017, Nutrients 9, 1073; Buhrmann et al., 2020, Biomedicines 8, 236;
Answer:
AS suggested we added more detail regarding migration into and introduced the references, in the introduction (lines 43- 48) and discussion (lines 398- 409) and sections.
Please write in the whole paper instead Integrin beta-1 or β-1: Integrin β1, α-6: α6 etc.
Answer:
Corrections have been performed
In Figure 4B: control protein is missing.
In Figure 6A-B: control protein is missing.
Answer:
We recognize controls are very important and we have included them all over the assays. Since figure 4 and 6 concern protein immunoprecipitates and respective fractions (cleared proteins from immunoprecipitate), using controls as beta tubulin may not be suitable as it would not appears in the immunoprecipiates. In these cases, we used exactly the same amount of protein for each experiment, i.e. 100 ug and all the resulting immunoprecipitated and cleared fractions were loaded on the gel. We realized this strategy was not clear in the text, so in this version, we add that information at the Material and Methods section (point 4.5) and also the amount of protein to the figure legends.
The authors should discuss the limitations of the study.
Answer:
In this new version, we have added a sentence referring to some limitations of the work. Please read the discussion section highlighted in yellow.
The whole blot of raw data of western blot must be shown.
Answer:
Please find a document showing the raw data of the Western blots.

Reviewer 2 Report
In this research paper, the authors have investigated how E-selectin and its carbohydrate ligand sialyl Lewis X antigen plays a role in helping to bind circulating tumor cells to the endothelium and increase metastasis. They characterized SW620 cells tranfected with fucosyltransferase 6 (FUT6) responsible for the synthesis of sialyl Lewis X antigen and demonstrated that these cells express high levels of sialyl Lewis X antigen. Their results showed that, these engineered cells (SW620FUT6) showed increased cell migration ability. Using, a variety of techniques, such as spectrophotometry, flow cytometry and Western Blot, they identified neural adhesion molecule L1 (L1CAM) as the most prominent E-selectin ligand. This research paper provides new insights on the role of L1CAM along with E-selectin in metastasis.
The research paper well written and the results presented supports conclusions. Its finding suggests a novel role for L1CA1 in tumor invasion and metastasis.
Minor comments:
The authors can add a conclusion section to highlight the findings after the discussion section. This section can be inserted after line 445.
Author Response
Thank you for reviewing the manuscript entitled L1CAM as an E-selectin ligand in colon cancer by Deschepper et al that has been submitted for consideration in the International Journal of Molecular Sciences
We have carefully considered the comments of the peer reviewers and have revised the manuscript accordingly. The changes performed in the revised version can be seen as highlighted text. We are honestly grateful to you, whose careful reading and suggestions have helped us to improve the manuscript.
We sincerely expect that you find this new version of the manuscript now acceptable.
Minor comments:
The authors can add a conclusion section to highlight the findings after the discussion section. This section can be inserted after line 445.
Answer:
Thank you for this comment. As suggested we adapted last paragraph for a conclusion section within the discussion. Here we introduced some limitation to the work and potential impact.
